# Dysprosium Substituted Ce:YIG Thin Films for Temperature Insensitive Integrated Optical Isolator Applications

**DOI:** 10.3390/ma15051691

**Published:** 2022-02-24

**Authors:** Zixuan Wei, Wei Yan, Jun Qin, Longjiang Deng, Lei Bi

**Affiliations:** National Engineering Research Center of Electromagnetic Radiation Control Materials, University of Electronic Science and Technology of China, Chengdu 610054, China; zxwei@std.uestc.edu.cn (Z.W.); qinjun@uestc.edu.cn (J.Q.); denglj@uestc.edu.cn (L.D.)

**Keywords:** magneto-optical materials, faraday rotation, temperature insensitivity

## Abstract

Magneto-optical isolators are key components in photonic systems. Despite the progress of silicon-integrated optical isolators, the Faraday rotation of silicon-integrated magneto-optical materials, such as cerium-doped yttrium iron garnet (Ce:YIG), show a strong temperature dependence, limiting the temperature range for integrated nonreciprocal photonic device applications. In this work, we report dysprosium substituted Ce:YIG thin films (Dy_2_Ce_1_Fe_5_O_12_, Dy:CeIG) showing a low temperature coefficient of Faraday rotation. A temperature insensitive range of the Faraday rotation is observed in between 25 °C to 70 °C for this material, compared to 20% variation of the Faraday rotation in Ce:YIG thin films. A Dy:CeIG based temperature insensitive silicon-integrated optical isolator operating in the temperature range of 23 °C to 70 °C is experimentally demonstrated.

## 1. Introduction

Integrated magneto-optical (MO) isolators and circulators are key components for silicon-integrated photonic circuits (PICs) [1,2,3]. At present, rare earth-doped yttrium iron garnet (RIG) is the most widely-used magneto-optical material in integrated MO devices. In the past decade, significant progress has been achieved in bonding or deposition of MO garnet thin films on silicon, such as cerium-doped yttrium iron garnet thin films (Ce:YIG). Silicon-integrated MO nonreciprocal photonic devices including optical isolators [4,5,6], circulators [1,7], modulators [8], and optical switches [9] have been developed, showing a high isolation ratio, low insertion loss, and fast switching/modulation speed. For practical applications, on-shelf bulk optical isolators or circulators using bismuth-doped rare earth iron garnet materials usually show a high performance in the temperature range from −20 °C to 70 °C. However, Ce:YIG show a stronger temperature dependence of the Faraday rotation, leading to large drift of the operation wavelength as a function of temperature [10,11]. This problem is particularly prominent for phase-sensitive devices such as the Mach–Zehnder interferometer (MZI) based optical isolators, which influences both the isolation ratio and insertion loss, resulting in a narrow operation temperature range.

Several previous works were carried out to address the temperature stability problem of Ce:YIG. Enno et al. studied the Faraday rotation angle of epitaxial Ce:YIG thin films from −175 °C to +25 °C [12]. The Faraday rotation monotonically decrease with increasing temperature. The Faraday rotation shows temperature dependence of about −20 deg·cm^−1^·K^−1^ in this temperature range. Huang et al. characterized silicon-integrated ring resonator optical isolators based on bonded Ce:YIG epitaxial thin films. They observed that the nonreciprocal-phase shift (NRPS) effect of Ce:YIG decreases when the material is heated up by an integrated electromagnet, which results in a reduced bandwidth and isolation ratio [13]. To solve this problem, Furuya et al. designed MZI optical isolators with a special designed length of the reciprocal phase shift (RPS) waveguide. The temperature dependence of the RPS and NRPS waveguide cancels out, results in an almost temperature insensitive transmission spectrum for the backward propagation direction in a temperature range of 20–60 °C [10]. However, the device still suffers from temperature dependence for forward propagation, leading to higher insertion loss when temperature is changed. A similar result is also observed in SiN-based optical isolators reported in our previous work [11]. Due to the low thermo-optic coefficient of SiN, the isolators maintained a relatively stable operation wavelength, showing a small wavelength drift within 4 nm in a temperature range of 20–70 °C. However, the device still shows a deteriorated performance due to the temperature dependence of the MO effect. Fundamentally, these problems originated from the temperature dependence of the Faraday effect of the MO material, which cannot be fully resolved by the device design. Therefore, exploring new MO materials with a higher temperature stability is important for solving temperature stability issues.

In this paper, we report a new MO material system: Dysprosium-substituted Ce:YIG (Dy:CeIG), showing a lower temperature dependence of the Faraday rotation compared to Ce:YIG films. Thanks to the negative temperature coefficient of the Faraday rotation of the Dy_3_Fe_5_O_12_ material [14,15,16], the Dy_2_Ce_1_Fe_5_O_12_ deposited on silicon shows a flat temperature dependence plateau with less than ±5% Faraday rotation change in the temperature range from 20 °C to 70 °C, in contrast to the almost 20% decrease of the Faraday rotation of Ce:YIG of the same temperature range. We also fabricated integrated MZI isolators on silicon using both materials, demonstrating higher stability of the Dy:CeIG based device. The good temperature stability of Dy:CeIG makes it a promising material for silicon-integrated nonreciprocal photonic device applications.

## 2. Materials and Methods

### 2.1. Material Fabrication and Characterization

Using the two-step deposition method of our previous study, we deposited 100 nm of thick Ce:YIG with 50 nm of thick YIG and 100 nm of thick Dy:CeIG with 50 nm of thick YIG, respectively, on (100) Si by pulsed laser deposition (PLD) [2,17]. Then the films were subsequently rapid thermal annealed at 850 °C under an oxygen partial pressure of 2 Torr for 3 min. The crystal structure was characterized by X-ray diffraction (Rigaku Ultima IV) with a Cu-K_α_ radiation source. The XRD patterns were collected over the 2θ range of 25–40° with a step size of 0.02°. Room-temperature in-plane and out-of-plane magnetic hysteresis loops were measured by vibrating sample magnetometry (VSM). The MO hysteresis loops were measured by using a custom-built Faraday effect characterization system with field and light propagation directions out of plane [17]. The maximum of the magnetic field is 4800 Gs. The sample holder of the Faraday effect characterization system is a tunable heater which has a heating range from room temperature (RT) to 120 °C.

### 2.2. Device Preparation and Characterization

Use the same methods in Section 2.1, 86 nm of Ce:YIG with 50 nm of YIG thin films and 180 nm of Dy:CeIG with 50 nm of YIG-thin films were deposited, respectively, on the top of SiN waveguides as discussed in [11]. The devices were characterized on a polarization-maintaining, fiber-butt-coupled system. The transmission spectrums were obtained from the same setup of our previous work [11]. An in-plane magnetic field (1000 Gs) was applied perpendicular to the propagation direction of light to saturate the magnetization of MO-thin films. Forward and backward transmittance between the input and output ports is given by [6]:(1)$$T_{for,back}=\cos^{2}\left(\frac{\theta_{RPS}\mp \theta_{NRPS}}{2}\right)$$
where *θ_RPS_* is the reciprocal phase shift (RPS) of the SiN waveguides, and *θ_NRPS_* is the nonreciprocal phase shift (NRPS) of the MO/SiN waveguides, which is linearly related to the Faraday rotation *θ_F_* [7,18]:(2)$$\theta_{NRPS}=\Delta \beta \cdot L_{NRPS}=L_{NRPS}\cdot \frac{2\beta }{\omega \varepsilon_{0}N}\iint \frac{\lambda \theta_{F}}{\pi n_{0}^{3}}H_{x}\partial_{y}H_{x}dxdy$$
where *ω* is the angular frequency, Δ*β* is the propagation constants difference between forward and backward propagation light, *β* is the propagation of the fundamental TM mode, *N* is the power flux along the z direction, *n*_0_ is the refractive index of the MO material [7], and *H_x_* is the magnetic field along the x direction.

From the measured NRPS, the Faraday rotation of the MO films can be calculated using the following equations:(3)$$\frac{\Theta_{f}(T)}{\Theta_{f}^{RT}}=\frac{\theta_{NRPS}(T)}{\theta_{NRPS}^{RT}}$$
where $\Theta_{f}^{RT}$ is the Faraday rotation angle of MO films at room temperature. $\theta_{NRPS}^{RT}$ is the NRPS at room temperature.

We consider the Faraday rotation is dispersionless in the measured wavelength range, which is a good approximation based on previous measurements [17]. Both devices are characterized under a temperature range from 23 °C (RT) to 100 °C, with a step size of 10 °C.

## 3. Results and Discussion

### 3.1. Material Characterization

Figure 1a shows the X-ray diffraction patterns of Dy:CeIG and Ce:YIG thin films. Both films show well-crystallized diffraction peaks of the polycrystalline garnet phase. The peak at 2θ = 33.1° originated from the (200) planes of the silicon substrate. Due to the ionic radius of Dy^3+^ ions (1.083 Å) being larger than Y^3+^ ions (1.02 Å), the lattice constant of Dy:CeIG is larger than that of Ce:YIG. The calculated lattice constants of Dy:CeIG and Ce:YIG are 12.42 Å and 12.37 Å, respectively, consistent with previous reports [17]. Figure 1b shows the room temperature (300 K) magnetic hysteresis loops of Dy:CeIG. The films show a similar saturation magnetization field for in-plane and out-of-plane magnetization directions, which is attributed to the competition between magnetoelastic anisotropy and shape anisotropy [17]. The saturation magnetization of Dy:CeIG thin film is 111 emu/cm^3^, comparable to our previous report [17].

Next, we characterized the Faraday rotation hysteresis loops of Dy:CeIG and Ce:YIG thin films in the temperature range of 25 °C to 120 °C, as shown in Figure 2a,b. We used the wavelength of 1310 nm for higher Faraday rotation angles and better visibility of the temperature dependence. The temperature dependence of the Faraday rotation can be quantified by the temperature coefficient:(4)$$\xi =\frac{1}{\theta_{f}}\cdot {\left(\frac{d\theta_{f}}{dT}\right)}_{T}(cm^{-1}\cdot K^{-1}).$$

For Ce:YIG, the saturation Faraday rotation angle reaches −6950 deg/cm at 25 °C. As the temperature rises, this value declines to −5730 deg/cm at 70 °C and −4500 deg/cm at 120 °C, showing a monotonic decrease in the measured temperature range, with a temperature coefficient of 3.7 × 10^−3^ cm^−1^·K^−1^. Whereas for Dy:CeIG, a lower temperature dependence of the Faraday rotation is observed. For the temperature range of 25 °C to 70 °C, the Faraday rotation angle of Dy:CeIG thin film decreases from −4250 deg/cm to −3800 deg/cm. When the temperature continues to rise, the Faraday rotation of the film decreases to −3200 deg/cm at 120 °C. A lower temperature coefficient of 2.6 × 10^−3^ deg·cm^−1^·K^−1^ is demonstrated in Dy:CeIG. The Faraday rotation value normalized to a room temperature value is summarized in Figure 2c,d for both materials. The error bar represents the standard deviation from three consecutive measurements at the same temperature. We notice the Faraday rotation of Ce:YIG decreases to almost 65% at 120 °C compared to its room temperature value, whereas for Dy:CeIG, it decreases to about 80%. In particular, for the temperature range of 70 °C and lower, which is the typical application temperature range for optical isolators, Dy:CeIG shows a much higher temperature stability compared to Ce:YIG.

### 3.2. Device Characterization

We further fabricated MZI isolator devices to measure the temperature stability of Dy:CeIG and Ce:YIG films [11]. Figure 3a,b shows the device structure. Details of the device design and fabrication can be found in our previous reports [11].

Figure 3c,d shows the transmission spectra of the TM-mode SiN optical isolator with Ce:YIG and Dy:CeIG as the MO materials. Clear one-way transmission is observed on both devices in the measurement temperature range. The forward and backward propagation spectra moved close to each other with an increasing temperature due to the lower NRPS and Faraday effects. For the Ce:YIG sample, the spectra shifted for Δλ_1_ − Δλ_2_ = 5.963 nm from 23 °C to 70 °C, as shown in Figure 3c. They further shifted for Δλ_1_ − Δλ_3_ = 9.732 nm from 70 °C to 100 °C. For the Dy:CeIG sample, the spectra showed less wavelength drift compared to Ce:YIG, which is Δλ_1_ − Δλ_2_ = 2.218 nm for the temperature range from 23 °C to 70 °C and Δλ_1_ − Δλ_3_ = 7.788 nm for the temperature range from 70 °C to 100 °C, respectively. Therefore the Dy:CeIG sample shows better temperature stability compared to Ce:YIG, especially for temperatures below 70 °C. Based on the transmission spectra of the devices at different temperatures, we calculated the Faraday rotation angle of Ce:YIG and Dy:CeIG at each temperature following Equation (3). Figure 3e,f shows the calculated Faraday rotation angle versus temperature for both materials at the measured wavelength range. The room temperature Faraday rotation angle at the 1550 nm wavelength of Ce:YIG and Dy:CeIG are −3500 deg/cm and −2000 deg/cm, respectively. For Ce:YIG, when the temperature increased from 23 °C to 100 °C, the Faraday rotation angle decreased from 3500 deg/cm to 2348 deg/cm, which was only 67% of the initial value, consistent with Figure 2c. For Dy:CeIG, the Faraday rotation angle increased to −2050 deg/cm up to 40 °C, then decreased to −1890 deg/cm at 70 °C, showing only a 5% change. The increase of the Faraday rotation below 40 °C is possibly due to the increase of the magnetization of this material at this temperature range, considering a compensation temperature of 225 K in Dy_3_Fe_5_O_12_ [19,20]. When the temperature continues to rise, the Faraday rotation angle of Dy:CeIG thin film decreased to –1628 deg/cm at 100 °C, which is 81.4% of the initial value. In terms of device performance, a 5% Faraday rotation variation leads to spectra shift of 2.5 nm for the current device structure. For Ce:YIG based devices, this spectra shift will take place as the temperature increases from 23 °C to 35 °C, whereas this temperature range is up to 23 °C to 100 °C for Dy:CeIG. Our observation demonstrates that Dy:CeIG based integrated optical isolators show higher temperature stability in the temperature range of 23 °C to 70 °C.

## 4. Conclusions

In summary, we demonstrated a temperature stable magneto-optical material Dy_2_Ce_1_Fe_5_O_12_ for silicon-integrated nonreciprocal photonic device applications. By replacing Y^3+^ ions with Dy^3+^ ions in Ce:YIG, we demonstrate less than a ±5% variation of the thin film Faraday rotation from the temperature of 25 °C to 70 °C, compared to ~20% in Ce:YIG. Integrated-MO isolators consist of Dy:CeIG thin films show better temperature stability compared to Ce:YIG based devices in the temperature range of 23 °C to 70 °C, which is highly desired for practical applications.

## Figures and Tables

**Figure 1 materials-15-01691-f001:**
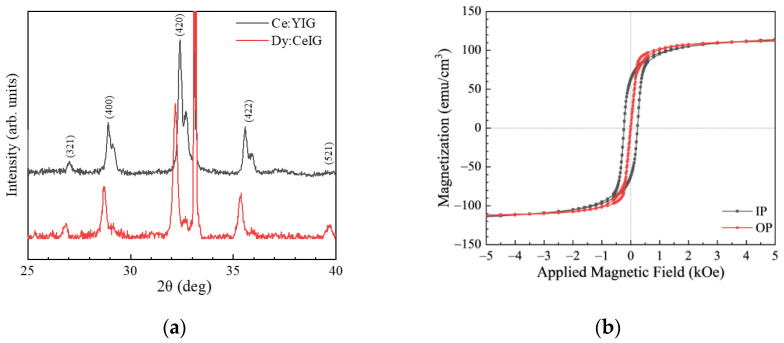
(**a**) X-ray diffraction patterns of Dy:CeIG and Ce:YIG thin films with a bottom yttrium iron garnet (YIG) seed layer of 50 nm in a 2θ range from 25° to 40°. (**b**) Room temperature in-plane and out-of-plane magnetization hysteresis loops for Dy:CeIG thin films.

**Figure 2 materials-15-01691-f002:**
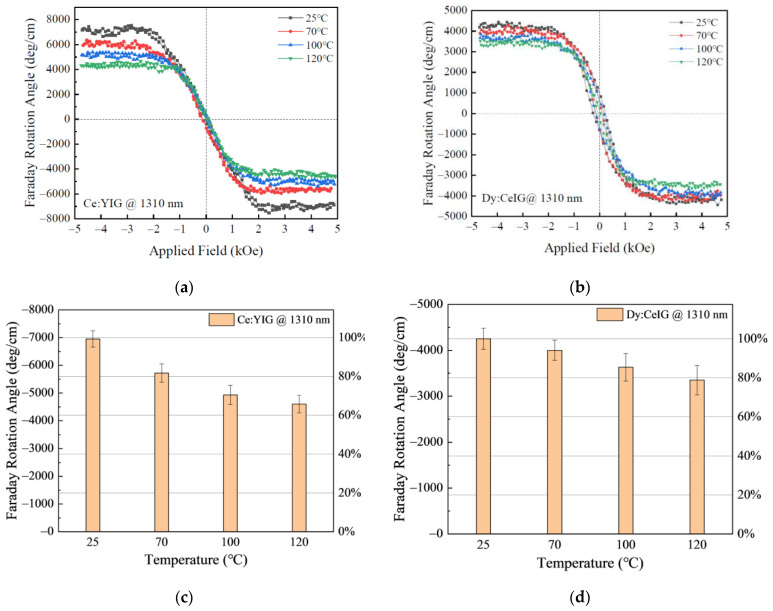
Temperature dependence of the Faraday rotation (FR) hysteresis loops at 1310 nm wavelength of (**a**) Ce:YIG and (**b**) Dy:CeIG thin films on silicon. Curves of different colors indicate different temperatures. Faraday rotation angle of both films as functions of temperature: (**c**) Ce:YIG and (**d**) Dy:CeIG. The error bar is the standard deviation from three consecutive measurements at the same temperature.

**Figure 3 materials-15-01691-f003:**
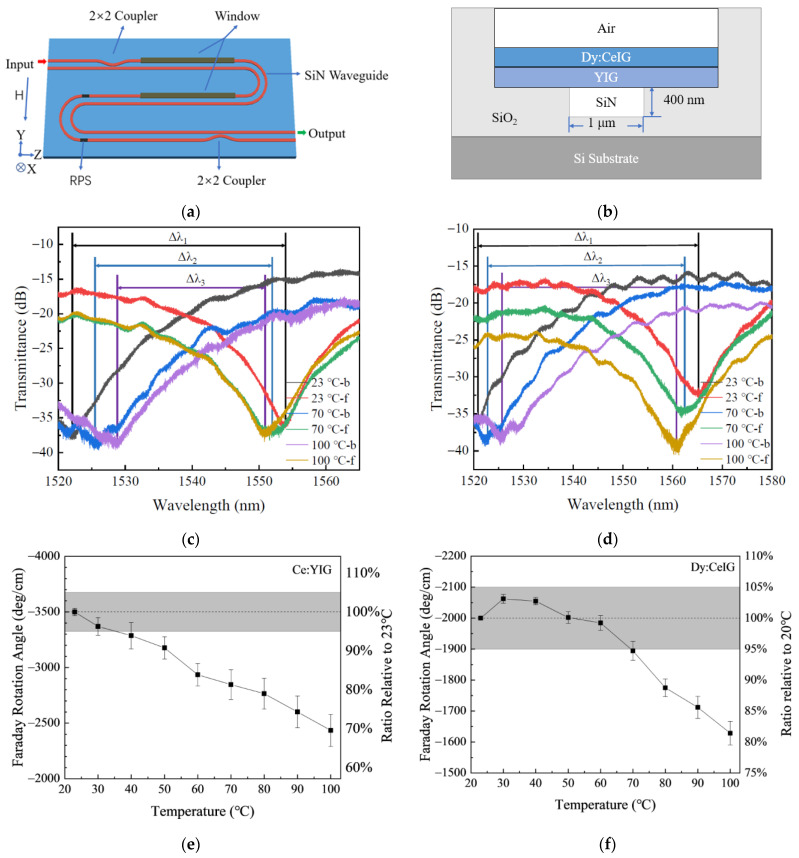
(**a**) Schematics of integrated magneto-optical isolators on SiN. (**b**) Cross section of MO/SiN waveguide. Transmission spectra of SiN optical isolators under different device temperatures of 23 °C, 80 °C, and 100 °C, respectively for (**c**) Ce:YIG and (**d**) Dy:CeIG. “f” and “b” correspond to forward and backward propagation light. Faraday rotation angle versus temperature resulting from transmission spectra, respectively for (**e**) Ce:YIG and (**f**) Dy:CeIG. The error bar is the standard deviation from three consecutive measurements of the transmission spectrum.

## Data Availability

The data presented in this study are available from the corresponding authors upon reasonable request.

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
