# Peer review of "Dysprosium Substituted Ce:YIG Thin Films for Temperature Insensitive Integrated Optical Isolator Applications"

_materials, 2022, doi:10.3390/ma15051691_

Round 1
Reviewer 1 Report
The article "Dysprosium substituted Ce:YIG thin films for temperature in-2 sensitive integrated optical isolator applications contains the results" presents a temperature stable magneto-optical material Dy2Ce1Fe5O12 for silicon integrated nonreciprocal photonic device applications. demonstrate a temperature stable magneto-optical material Dy2Ce1Fe5O12 for silicon integrated nonreciprocal photonic device applications. By replacing Y3+ ions with Dy3+ ions in Ce:YIG, authors demonstrate less than ±5% variation of the thin film Faraday rotation from the temperature of 25 ℃ to 70 ℃, compared to ~20% in Ce:YIG. It is shown that integrated MO isolators consist of Dy:CeIG thin films show better temperature stability compared to Ce:YIG based devices in the temperature range of 23 ℃ to 70 ℃.
The article contains many original results of undeniable practical significance. It will be of interest to a wide range of researchers involved in magneto-optical resonators.
There are several questions about the content of the article.
1) It is not indicated in the introduction whether any alternative to thin films of yttrium iron garnet is currently being considered for use as magneto-optical resonators.
2) How will the temperature dependence of the Faraday rotation hysteresis loops of Dy:CeYIG thin films change when moving to another wavelength range? Is there any dependence on the wavelength?
3) How can one explain the increase in the value of the Faraday rotation angle in the range of 30-40 degrees for Dy:CeYIG (Fig. 3f), which is absent for Ce:YIG (Fig. 3f)?
Author Response
Response to Reviewer 1 Comments
Reviewer #1 (Comments to the Author):
The article "Dysprosium substituted Ce:YIG thin films for temperature in-2 sensitive integrated optical isolator applications contains the results" presents a temperature stable magneto-optical material Dy2Ce1Fe5O12 for silicon integrated nonreciprocal photonic device applications. demonstrate a temperature stable magneto-optical material Dy2Ce1Fe5O12 for silicon integrated nonreciprocal photonic device applications. By replacing Y3+ ions with Dy3+ ions in Ce:YIG, authors demonstrate less than ±5% variation of the thin film Faraday rotation from the temperature of 25 ℃ to 70 ℃, compared to ~20% in Ce:YIG. It is shown that integrated MO isolators consist of Dy:CeIG thin films show better temperature stability compared to Ce:YIG based devices in the temperature range of 23 ℃ to 70 ℃.
The article contains many original results of undeniable practical significance. It will be of interest to a wide range of researchers involved in magneto-optical resonators.
Response: We appreciate the reviewer’s comments to our work. We answer the reviewer’s questions as follows.
Point 1: It is not indicated in the introduction whether any alternative to thin films of yttrium iron garnet is currently being considered for use as magneto-optical resonators.
Response 1: Thanks for the commerts. To the best of our knowledge, rare earth doped yttrium iron garnet is still the main material used for magneto-optical devices. They include bismuth doped YIG, bismuth, terbium doped YIG and cerium doped YIG. Other materials such as Wely semimatels are still under theoretical study[1]. We have added comments to these materials in the manuscript.
Revisions: Page 1, line 21, added “At present, rare earth doped yttrium iron garnet (RIG) is the most widely used magneto-optical material in integrated MO devices.”
Point 2: How will the temperature dependence of the Faraday rotation hysteresis loops of Dy:CeYIG thin films change when moving to another wavelength range? Is there any dependence on the wavelength?
Response 2: Thanks for the commerts. Absoultly there is also temperature dependence of the Faraday rotation when shift to other wavelengths. However, the transparency window of this material is in the 1550 nm wavelength range. When moving to shorter wavelengths, the absorption of this material increases sharply, making them less practical for photonic device applications.
Point 3: How can one explain the increase in the value of the Faraday rotation angle in the range of 30-40 degrees for Dy:CeYIG (Fig. 3f), which is absent for Ce:YIG (Fig. 3f)?
Response 3: Thanks for the commerts. The saturation magnetization of rare-earth doped YIG affects the Faraday rotation angle. Sayetat et al. and Ostorero et al. previously reported that the compensation temperature of DyIG is Tcomp = 225 K [2,3], which could decrease the Faraday rotation angle of Dy:CeIG in this work at the temperature below 300 K. Therefore, a higher Faraday rotation may be related to the increase of the saturation magnetization of this material in this temperature range.
Revisions:Page 5, line 170-173, added “The increase of the Faraday rotation below 40 ℃ is possibly due to the increase of the magnetization of this material at this temperature range, considering a compensation temperature of 225 K in Dy3Fe5O12 [19,20].”.
References:
- Okamura, Y.; Minami, S.; Kato, Y.; Fujishiro, Y.; Kaneko, Y.; Ikeda, J.; Muramoto, J.; Kaneko, R.; Ueda, K.; Kocsis, V.; et al. Giant magneto-optical responses in magnetic Weyl semimetal Co3Sn2S2. Nat Commun 2020, 11, 4619, doi:10.1038/s41467-020-18470-0.
- Sayetat, F. Huge magnetostriction in Tb3Fe5O12, Dy3Fe5O12, Ho3Fe5O12, Er3Fe5O12 garnets. Journal of magnetism and magnetic materials 1986, 58, 334-346.
- Ostorero, J.; Escorne, M.; Pecheron‐Guegan, A.; Soulette, F.; Le Gall, H. Dy3Fe5O12 garnet thin films grown from sputtering of metallic targets. Journal of Applied Physics 1994, 75, 6103-6105.
Reviewer 2 Report
This manuscript relates to the effect of dysprosium substitution in Ce:YIG thin films and in particular to the temperature dependance of their Faraday rotation. It is in general well written and the work deserves publication. There are just a few issues that the authors should take into account for further improving their paper prior to acceptance:
1) For several references, the text "Error! Reference source not found" appears. Please check with the editor if this can be fixed.
2) In equation (2) and the following line, there are inconcistencies in the use of capital letters for the Faraday rotation angles.
3) Line 111. "X-ray diffraction patterns", not "spectra".
4) Figures 2 (c) and (d). While the Faraday rotation angle appears to decrease more or less linearly with temperature in the Ce:YIG film (as far as can be inferred from the non-linear horizontal scale), the decrease is not taking place at the same rate in the Dy:Ce:YIG film. What causes this different behavior?
5) Figures 3 (a) and (b): It could be good to use the same horizontal scale for a clearer comparison.
Author Response
Response to Reviewer 2 Comments
Reviewer #2 (Comments to the Author):
This manuscript relates to the effect of dysprosium substitution in Ce:YIG thin films and in particular to the temperature dependance of their Faraday rotation. It is in general well written and the work deserves publication.
Response: Thanks for your recognition of our work. We have answered the questions as follows.
Point 1: For several references, the text "Error! Reference source not found" appears. Please check with the editor if this can be fixed.
Response 1: Thanks for correcting this. We have updated the references and revised this in the context.
Revisions: Page 1, line 21, revised to “for silicon integrated photonic circuits (PICs) [1-3].”. Page 1, line 24, revised to “including optical isolators [4-6]”. Page 1, line 41, revised to “which results in reduced bandwidth and isolation ratio [13].”. Page 1, line 45, revised to “in a temperature range of 20-60 ℃ [10].”. Page 5, line 148, revised to “measure the temperature stability of Dy:CeIG and Ce:YIG films[11].”.
Point 2: In equation (2) and the following line, there are inconcistencies in the use of capital letters for the Faraday rotation angles.
Response 2: Thanks for correcting this. We have revised this in the context.
Revisions: Page 3, line 98, revised to “where is the Faraday rotation angle of MO films at room temperature. is the NRPS at room temperature.”
Point 3: Line 111. "X-ray diffraction patterns", not "spectra".
Response 3: Thanks for correcting this. “X-ray diffraction spectra” has been revised into “X-ray diffraction patterns”.
Revisions: Page 3, line 109, revised to “X-ray diffraction patterns”.
Point 4: Figures 2 (c) and (d). While the Faraday rotation angle appears to decrease more or less linearly with temperature in the Ce:YIG film (as far as can be inferred from the non-linear horizontal scale), the decrease is not taking place at the same rate in the Dy:Ce:YIG film. What causes this different behavior?
Response 4: Thanks for the commerts. The saturation magnetization of rare-earth doped YIG affects the Faraday rotation angle. Sayetat et al. and Ostorero et al. previously reported that the compensation temperature of DyIG is Tcomp = 225 K [1,2], which could decrease the Faraday rotation angle of Dy:CeIG in this work at the temperature below 300 K. Therefore, a higher Faraday rotation may be related to the increase of the saturation magnetization of this material in this temperature range.
Revisions: Page 5, line 170-173, added “The increase of the Faraday rotation below 40 ℃ is possibly due to the increase of the magnetization of this material at this temperature range, considering a compensation temperature of 225 K in Dy3Fe5O12 [19,20].”.
Point 5: Figures 3 (a) and (b): It could be good to use the same horizontal scale for a clearer comparison.
Response 5: Thanks for pointing this out. We have revised the Figure 3.
Revisions: Page 6, revised the horizontal scale of Figure 3 (c) and updated Figure 3 (c).
References:
- Sayetat, F. Huge magnetostriction in Tb3Fe5O12, Dy3Fe5O12, Ho3Fe5O12, Er3Fe5O12 Journal of magnetism and magnetic materials 1986, 58, 334-346.
- Ostorero, J.; Escorne, M.; Pecheron‐Guegan, A.; Soulette, F.; Le Gall, H. Dy3Fe5O12 garnet thin films grown from sputtering of metallic targets. Journal of Applied Physics 1994, 75, 6103-6105.